# SKI-1/S1P Facilitates SARS-CoV-2 Spike Induced Cell-to-Cell Fusion via Activation of SREBP-2 and Metalloproteases, Whereas PCSK9 Enhances the Degradation of ACE2

**DOI:** 10.3390/v15020360

**Published:** 2023-01-27

**Authors:** Rachid Essalmani, Ursula Andréo, Alexandra Evagelidis, Maïlys Le Dévéhat, Oscar Henrique Pereira Ramos, Carole Fruchart Gaillard, Delia Susan-Resiga, Éric A. Cohen, Nabil G. Seidah

**Affiliations:** 1Laboratory of Biochemical Neuroendocrinology, Montreal Clinical Research Institute (IRCM), Université de Montréal, Montreal, QC H2W 1R7, Canada; 2Département Médicaments et Technologies pour la Santé (DMTS), Université Paris-Saclay, CEA, INRAE, SI-MoS, 91191 Gif-sur-Yvette, France; 3Laboratory of Human Retrovirology, Montreal Clinical Research Institute (IRCM), Université de Montréal, 110 Pine Ave West, Montreal, QC H2W 1R7, Canada; 4Department of Microbiology, Infectiology and Immunology, Université de Montréal, Montreal, QC H3C 3J7, Canada

**Keywords:** cell-to-cell fusion, proprotein convertases, metalloproteases, mutagenesis, PCSK9, protease inhibitors, SARS-CoV-2, shedding, SKI-1/S1P, SREBP-2

## Abstract

Proprotein convertases activate various envelope glycoproteins and participate in cellular entry of many viruses. We recently showed that the convertase furin is critical for the infectivity of SARS-CoV-2, which requires cleavage of its spike protein (S) at two sites: S1/S2 and S2′. This study investigates the implication of the two cholesterol-regulating convertases SKI-1 and PCSK9 in SARS-CoV-2 entry. The assays used were cell-to-cell fusion in HeLa cells and pseudoparticle entry into Calu-3 cells. SKI-1 increased cell-to-cell fusion by enhancing the activation of SREBP-2, whereas PCSK9 reduced cell-to-cell fusion by promoting the cellular degradation of ACE2. SKI-1 activity led to enhanced S2′ formation, which was attributed to increased metalloprotease activity as a response to enhanced cholesterol levels via activated SREBP-2. However, high metalloprotease activity resulted in the shedding of S2′ into a new C-terminal fragment (S2″), leading to reduced cell-to-cell fusion. Indeed, S-mutants that increase S2″ formation abolished S2′ and cell-to-cell fusion, as well as pseudoparticle entry, indicating that the formation of S2″ prevents SARS-CoV-2 cell-to-cell fusion and entry. We next demonstrated that PCSK9 enhanced the cellular degradation of ACE2, thereby reducing cell-to-cell fusion. However, different from the LDLR, a canonical target of PCSK9, the C-terminal CHRD domain of PCSK9 is dispensable for the PCSK9-induced degradation of ACE2. Molecular modeling suggested the binding of ACE2 to the Pro/Catalytic domains of mature PCSK9. Thus, both cholesterol-regulating convertases SKI-1 and PCSK9 can modulate SARS-CoV-2 entry via two independent mechanisms.

## 1. Introduction

Severe acute respiratory syndrome coronavirus 2 (SARS-CoV-2) is responsible for the global pandemic of COVID-19 infecting over 675 million individuals and counting since 2019, resulting in ~1% lethality (https://www.worldometers.info/coronavirus/ accessed on 25 January 2023). Despite the tremendous success of vaccines in preventing hospitalization, severe disease, and death, SARS-CoV-2 is still spreading, and numerous new variants continuously emerge that partially escape from immunization strategies [1,2].

Like envelope glycoproteins of many infectious viruses, the secretory type-I membrane-bound spike (S) protein of SARS-CoV-2 is synthesized as a precursor (proS) that undergoes post-translational cleavages by host cell proteases at specific sites to allow viral entry. The proS monomer of 1273 residues is first processed at an S1/S2 cleavage site. Unlike SARS-CoV-1, the S protein of SARS-CoV-2 exhibits an insertion of four critical amino acids (PRRA) at the S1/S2 junction, forming a canonical PRRAR_685_↓ furin-like cleavage site [3]. Furin, the third member of the nine-membered proprotein convertase (PC) family [4], is implicated in the processing of many envelope glycoproteins and, therefore, actively regulates virus entry [5]. Since our first report, various studies have demonstrated the implication of furin in SARS-CoV-2 infection of human cell culture models in vitro [6,7,8,9] and in vivo in mice, hamsters, and ferrets [10,11]. Following S-protein priming, the S1 ectodomain undergoes a conformational change that exposes its receptor-binding domain (RBD) [12], which recognizes the angiotensin-converting enzyme 2 (ACE2) entry receptor [13]. The S2 subunit is responsible for the fusogenic activity of the S glycoprotein. Cleavage at the second proteolytic site (S2′) liberates the α-helical fusion peptide (FP) and two heptad repeat domains (HR1 and HR2) preceding the transmembrane domain (TM) and cytosolic tail, favoring fusion of viral and host cell membranes, and leading to virus entry. Fusion with host cells can occur either at the cell surface (pH-independent) or with internal membranes following endocytosis (pH-dependent) [14]. However, the cognate host cell proteases responsible for the S1/S2 and S2′ cleavages vary between coronaviruses and cell types [6,9,13,15,16,17,18,19], with furin being critical for the S1/S2 cleavage [9].

Metalloproteases, especially the sheddases ADAM10 and ADAM17 [20] have also been implicated in the S-protein processing into S2′ fragment and, hence, in SARS-CoV-2 cellular entry and syncytia formation [21,22]. Shedding activities of ADAMs were reported to be sensitive to membrane cholesterol content or distribution [23,24] and in vivo to be regulated by cholesterol and mevalonate [25]. Different from ADAM10, ADAM17 segregates into cholesterol-rich lipid rafts together with furin, a process that contributes to its maximal activation [26].

The convertase SKI-1 (also known as S1P), the eighth member of the PC-family, is involved in the cleavage of viruses recognizing the general motif R-X-Aliphatic-Z↓, where X is any residue except Pro and Cys, and Z is any aa (best Leu) except Val, Pro, Cys, or Glu and is most active in the cis/medial Golgi [4,5,27,28]. Interestingly, an important role of SKI-1 is to activate sterol regulatory element-binding protein (SREBP) transcription factors, which implicates SKI-1 in lipid and cholesterol metabolism [29,30]. A recent genome-wide scale CRISPR knockout screen uncovered SREBP signaling including SKI-1 and SREBP-2, among the essential pathways for infection by SARS-CoV-2 and three seasonal coronaviruses [31].

PCSK9 [32], the ninth member of the PC-family [4], is a major regulator of LDL-cholesterol (LDLc), since it escorts the cell surface receptor LDLR to endosomes/lysosomes for degradation, thereby enhancing circulating LDLc levels [33,34,35]. Lower levels of LDLc seem to be associated with reduced SARS-CoV-2 infections, since infected patients pretreated with the cholesterol-reducing “statins” exhibit reduced mortality and risk of COVID-19 hospitalization [36,37,38]. Indeed, SARS-CoV-2 requires cholesterol for viral entry and pathological syncytia formation [39]. Finally, it should be noted that human PCSK9 abundantly circulates in plasma and is also expressed in the vasculature of lungs, the primary infection site of SARS-CoV-2 [40]. In fact, non-infected patients with pulmonary arterial hypertension benefited from PCSK9 inhibition (DOI: https://www.researchsquare.com/article/rs-1965292/v1 accessed on 25 January 2023).

No report has yet been published on the potential implication of either SKI-1 or PCSK9 in SARS-CoV-2 spike processing and viral entry. Herein, we first show that in HeLa cells, enhanced SKI-1 activity potentiates cell-to-cell fusion, likely via the activation of SREBP-2, suggesting that a combination of furin and SKI-1 or metalloprotease inhibitors could be added to the therapeutic arsenal against SARS-CoV-2 infection. Our results suggest that high levels of metalloproteases and PCSK9 may protect against viral infection by enhancing the shedding of S-glycoprotein and the degradation of ACE2, respectively.

## 2. Materials and Methods

### 2.1. Plasmids

Doubly tagged (N-terminal hemagglutinin [HA] tag and C-terminal V5 tag) spike glycoprotein of SARS-CoV-2 (optimized sequence) and its mutants were cloned into the pIRES2-EGFP vector. Site-directed mutagenesis was achieved using a QuikChange kit (Stratagene, CA, USA) according to the manufacturer’s instructions. Plasmid pCI-NEO-hACE2 was received from D. W. Lambert (University of Leeds, Leeds, UK), and plasmid pIRES-NEO3-hTMPRSS2 was received from P. Jolicoeur (IRCM, Montreal, QC, Canada). The pHIV-1NL4-3 ΔEnv-NanoLuc construct was a kind gift from Paul Bieniasz.

### 2.2. Cell Culture and Transfection

Monolayers of HeLa, Hela-TZM-bl, and Calu-3 cells were cultured in 5% CO_2_ at 37 °C in DMEM (Wisent, St Bruno, QC, Canada) supplemented with 10% (*v*/*v*) FBS (Wisent, St Bruno, QC, Canada). HeLa cells were transfected with JetPrime transfection reagent according to the manufacturer’s instructions (Polyplus-transfection, VWR, New York, NY, USA). At 24 h post-transfection, the culture media were changed to serum-free, and the cells were incubated for an additional 24 h.

To generate HIV particles pseudotyped with SARS-CoV-2 S, HEK293T17 cells (600,000 cells plated in a 6-well vessel) were transfected with 1 μg of pHIV-1NLΔEnv-NanoLuc in the presence or absence of 0.3 μg of pIR-2019-nCoV-S V5 plasmids using JetPrime transfection reagent according to the manufacturer’s instructions (Polyplus-transfection, VWR, New York, NY, USA). Pseudovirions expressing nanoluciferase were collected at 24 h or 48 h post-transfection, respectively. Viral supernatants were clarified by centrifugation at 300× *g*, passed through a 0.45-μm-pore-size polyvinylidene fluoride (PVDF; Millipore, Oakville, ON, Canada) syringe filter (Millipore, Oakville, ON, Canada; SLGVR33RS), and aliquots were frozen at −80 °C. HIV particles lacking the SARS-CoV-2 S glycoprotein served as a negative control in all experiments. Production of pseudoparticles was quantified by ELISA using a p24 ELISA according to the manufacturer instruction (EXPRESS Bio, Toronto, ON, Canada).

### 2.3. Western Blots 

Cells were washed with phosphate-buffered saline (PBS) and then lysed in radioimmunoprecipitation assay (RIPA) buffer (1% Triton X-100, 150 mM NaCl, 5 mM EDTA, and 50 mM Tris [pH 7.5]) for 30 min at 4 °C. The cell lysates were collected after centrifugation at 14,000× *g* for 10 min. The proteins were separated on 8% Tricine gels by SDS-PAGE and transferred to a PVDF membrane (PerkinElmer). The proteins were revealed using a V5-monoclonal antibody (V5-mAb) (V2660; 1:5000; Invitrogen, Waltham, MA, USA), as well as antibodies against ACE2 (rabbit monoclonal antibody ab108252; 1:3000; Abcam, Toronto, ON, Canada), actin (rabbit polyclonal A2066; 1:5000; Sigma, Oakville, ON. Canada), or HA-horseradish peroxidase (HRP) (12-013-819; 1:3500; Roche, Laval, QC, Canada). The antigen-antibody complexes were visualized using appropriate HRP-conjugated secondary antibodies and an enhanced chemiluminescence kit (ECL; Amersham, Oakville, ON. Canada or Bio-Rad, Oakville, ON. Canada). Quantification of immunoreactive proteins was performed using Image Lab software (Bio-Rad, Oakville, ON. Canada) and their reported values are relative to β-actin.

### 2.4. Cell-to-Cell Fusion Assay

HeLa or HeLa TZM-bl cells were plated at 200,000 in 12-well plates. HeLa cells were transiently transfected with different constructs of SARS-CoV-2 spike or NL4.3-HIV Env (as control) or an empty vector (EV) and 0.2 μg of cytomegalovirus (CMV) Tat plasmid. HeLa TZM-bl cells were transfected with human ACE2. At 6 h post-transfection, media were replaced with fresh ones in absence or presence of various inhibitors, and 24 h later, the cells were detached with PBS-EDTA (1 μM). Different combinations of HeLa and HeLa TZM-bl cells were placed in coculture plates at a ratio of 1:1 for a total of 60,000 cells/well of a 96-well plate. After 18 to 24 h, the media were removed and 50 μL of cell lysis reagent was added in each well. Twenty microliters of the cell lysate were used for luciferase assay using 50 μL of Renilla luciferase reagent (Promega, Madison, WI, USA). Relative light units (RLU) were measured using a Promega GloMax plate reader and values were reported as fold increase over the RLU measured in co-culture of HeLa cells transfected with an empty vector with respective TZM-bl cells.

### 2.5. Pseudo-Virus Entry

Calu-3 cells (10,000 cells/well plated in a 96-well dish for 48 h) were transduced with 12 ng p24 equivalent of filtered pseudovirions overnight [9]. Target cells were pretreated with BOS-981 (1 μM) for 6 h before transduction. The overnight incubation with pseudovirions was performed in the presence of the inhibitors. Viral inoculum was removed, then fresh media were added, and the cells were cultured for up to 72 h. Upon removal of spent media, Calu-3 cells were gently washed twice with PBS and analyzed for nanoluciferase activity, respectively, using the Promega Nano-Glo luciferase system (Madison, WI, USA).

### 2.6. Inhibitor Treatment

At 24 h post transfection, cells were incubated for 16 h with DMSO (non-treated, NT), 3 μM of a selective cell-permeable furin-like inhibitor (BOS-981; kindly provided by Boston Pharmaceuticals [9]), 10 μM SKI-1 inhibitor or 1 μM metalloproteinase inhibitors: BB-94, ZLDI-8 (MedChemExpress, Princeton, NJ, USA), Phenanthroline (Abcam, Cambridge, UK), Merimastat (Sigma-Aldrich, St. Louis, MO, USA), and GM6001 (Chemicon, Burlington, ON, USA). The cells were washed with phosphate-buffered saline, lysed using radioimmunoprecipitation assay and then collected after centrifugation (14,000× *g*, 10 min) for further analysis.

### 2.7. HEK293 Cells Conditioned Media Production and Media Swap

PCSK9 conditioned media were produced by transfection of HEK293 cells with PCSK9 WT and its various mutant constructs. The media were collected 48 h post-transfection, centrifuged 20 min (300× *g* at 4 °C), aliquoted and stored at −80 °C until use. For media swap, HeLa cells were pre-incubated in serum-free medium for 1 h, followed by conditioned media swap for 18 h. The cells were then collected for further analysis.

### 2.8. Statistical Analysis

The difference between the control and the treated cells were evaluated by Student’s *t*-test. *p* values of 0.05 or lower were considered statistically significant (*, *p* < 0.05; **, *p* < 0.01; ***, *p* < 0.001, ****, *p* < 0.0001). All data are representative of at least three independent experiments. For the cell-to-cell fusion, representative data are shown as mean values (4 to 6 replicates) ± SD. The extent of fusion is estimated from the ratio between the relative luminescence units (RLU) measured for each condition.

### 2.9. Molecular Modeling of PCSK9/ACE2 Complex

PCSK9/ACE2 complex was obtained from the 10 best scored models constructed using GRAMM docking web server (https://gramm.compbio.ku.edu/ accessed on 25 January 2023) using PCSK9 structure (PDB:2P4E) as receptor and ACE2 dimeric structure (PDB:6M18) as ligand using default parameters except for the constraint of ACE2 R_697_ and E_699_ at the complex interface (confidence score: 10).

## 3. Results

### 3.1. SKI-1 Activity Enhances Spike-Induced Cell-Cell Fusion but Not Pseudoparticle Entry

SARS-CoV-2 can enter cells upon binding of the Spike glycoprotein to its receptor ACE2 by two routes: (1) a pH-independent plasma membrane pathway requiring the priming of S by furin cleavage at S1/S2 [3,6,7,8,9] followed by activation at S2′ by furin and/or TMPRSS2 [9,17,41] or metalloproteases such as ADAM10 and ADAM17 [21] (Figure 1A); (2) a pH-dependent endocytosis pathway involving the entry of the uncleaved S-ACE2 complexed particle into acidic early endosomes, where processing of S occurs by lysosomal enzymes such as cathepsins [42] (Figure 1A).

To investigate the potential regulation of SARS-CoV-2 fusion by SKI-1, we performed a cell-to-cell fusion assay based on the co-culture of donor Hela cells transiently expressing wild type (WT)-S and HIV-Tat with acceptor tat-driven luciferase reporter Hela TZ-Mbl cells, transiently expressing ACE2, as described in [9]. WT-S promotes fusion and syncytia formation leading to the transfer of Tat from Hela donor cell to the reporter TZM-bl cells driving Tat expression and, hence, luciferase activation (Figure 1B).

Upon overexpression of SKI-1 in either donor (expressing S) and/or acceptor (expressing ACE2) HeLa cells, we observed a significant ~1.6-fold fold enhanced cell-to-cell fusion (Figure 1C). To confirm that the observed enhanced fusion is due to an increase in SKI-1 protease activity, we performed similar experiments with an inactive SKI-1 (μSKI-1) mutated at its active site His_249_ into Ala [43]. This resulted in the abrogation of the effect of over-expressed SKI-1 (Figure 1D), confirming that the protease activity of SKI-1 in S-expressing donor cells as well as in ACE2-expressing acceptor cells is critical for its observed effect on S-directed fusion. Next, we inhibited the activity of endogenous SKI-1 during the co-culture of donor/acceptor HeLa cells by incubation with 10 µM PF429242 (Figure 1E), a cell permeable SKI-1 inhibitor [44], indicating that the lack of SKI-1 activity prevented cell-to-cell fusion. In contrast, using lung-derived human Calu-3 cells, we showed that 10 µM PF429242 treatment had no effect on pseudoviral particle entry (Figure 1F), indicating that SKI-1 might not play a major role in the pH-independent entry of SARS-CoV-2 in Calu-3-cells [9], likely because of the dominant presence of TMPRSS2 [6].

### 3.2. SKI-1 Activity Enhances the Generation of S2′, Favors a Cleavage at S2″ and the Shedding of a Long Form of S (S1_VL_)

We first tested whether in the presence of ACE2 the protease activity of SKI-1 that enhances cell-to-cell fusion (Figure 1) results in the direct cleavage of proS into S1/S2 or S2′ (Figure 2A). Thus, we co-expressed SKI-1 with doubly tagged [HA]-S-[V5] [9] in HeLa cells in absence or presence of BOS-981, a cell-permeable small molecule potent inhibitor of furin-like convertases [9]. The data show that in the presence of ACE2, proS is cleaved at S1/S2 by endogenous furin-like enzymes as reported [9], but we additionally observed the formation of an as yet unreported small membrane-bound C-terminal V5-positive ~11 kDa fragment of S herein called S2″ (Figure 2A,B). Overexpression of SKI-1 resulted in the enhanced generation of S2′ and S2″. The presence of BOS-981 eliminated S2 formation by furin (see EV, empty vector) as published [9], but still allowed S2′ and S2″ formation, and the secretion of the corresponding N-terminal HA-tagged ~245 kDa fragment S1_VL_ (Figure 2B). The data on S2′ were confirmed with S1/S2 mutants (µS1/S2 and µAS1/S2) that prevent furin cleavage [9] (Appendix A). To probe the possible direct SKI-1 cleavage of proS into S2′-like fragments, we scanned the protein sequence of proS around the bona fide S2′ site (PSKR_815_↓SF) [9]. Accordingly, only two possible SKI-1 cleavage sites that would fit the minimal best SKI-1 recognition motif R-X-L-Z [5] were found, namely RALT_768_-GI and RDLI_850_-CA. However, the R765A and R847A mutants of proS did not prevent the enhanced generation of S2′ by SKI-1 (Appendix A). We conclude that SKI-1 is not directly implicated in the cleavage of proS but may indirectly enhance such processing by activating other enzymes, such as the metalloproteases ADAM10 and ADAM17 reported to also cleave at S2′ [21], as we observed in the absence of furin activity upon BOS-981 inhibition (Figure 2B).

Consistently, the incubation of cells with various metalloprotease inhibitors resulted in an extensive reduction of S2″ formation (Figure 2C), supporting the notion that S2″ is generated by metalloproteases that are activated in the presence of SKI-1. Note that absence of metalloprotease activity did not significantly affect the levels of S2′, likely generated by endogenous furin in HeLa cells (Figure 2C). To validate the functionality of the cell-permeable SKI-1 inhibitor PF-429242 in HeLa cells [45,46], we selected proBrain Derived Neurotrophic Factor (proBDNF) as a substrate, since it was reported to be first cleaved by SKI-1 into a 28 kDa intermediate, which together with proBDNF are cleaved by furin to produce a 12.5 kDa mature BDNF [28]. Accordingly, co-expression of proBDNF with either SKI-1 or furin generated the expected products, but the incubation of cells with PF-429242 selectively eliminated the formation of the 28 kDa product, without affecting the generation of mature BDNF by furin (Appendix A). We next tested the effects of inhibitors of furin, metalloproteases, and SKI-1 on cell-to-cell fusion of HeLa cells. The data showed that inhibition of furin by BOS-981 [9], of metalloproteases by Batimastat (BB94) [47], and of SKI-1 by PF-429242 [45,46] reduced cell-to-cell fusion by 70–90% (Figure 2D). In addition, the combination of BOS-981 + BB-94 further reduced fusion up to 95%, but more significantly the combination of BOS-981 and PF-429242 effectively eliminated cell-to-cell fusion (Figure 2D). We conclude that in HeLa cells, SKI-1 activity indirectly increases endogenous metalloprotease activities that can enhance fusion (Figure 1C) and S2′ formation (Figure 2B).

### 3.3. Mevalonate and nSREBP-2 Mimic the SKI-1 Effect on Promoting Cell-to-Cell Fusion but Not Entry of Pseudoparticles

Among the best known functions of SKI-1 is the cleavage-activation of SREBP transcription factors [29], which when followed by cleavage by S2P would generate an N-terminal soluble nuclear form (nSREBP), e.g., nSREBP-2 that regulates the synthesis and cellular uptake of cholesterol by activating genes such as HMG-CoA reductase (HMGCR), HMG-CoA synthase (HMGCS), and mevalonate kinase (MVK) or LDL receptor (LDLR) [34,48]. Our data revealed that overexpression of either SKI-1 or nSREBP-2 in acceptor HeLa-ACE2 cells similarly enhanced cell-to-cell fusion by ~40–50% relative to ACE2 alone (Figure 3A). Since activation of SREBP-2 into nSREBP-2 enhances cholesterol synthesis via a mevalonate intermediate [49,50], we tested whether providing mevalonate directly to cells may also enhance cholesterol formation, remodel membranes, and act as a shortcut for SREBP-2 activation. Indeed, in presence of mevalonate, we observed a significant ~60% increase in cell-to-cell fusion relative to S-WT alone (Figure 3B). Consistently, with the absence of effect of SKI-1 inhibitors, mevalonate had no significant effect on pseudoparticle entry into Calu-3 cells (Figure 3C). These results indicate that SKI-1 enhances cell-to-cell fusion via enhanced formation of nSREBP-2, and that mevalonate has no significant effect on entry of pseudoparticles in Calu-3 cells, likely because of the presence of endogenous TMPRSS2 in these cells [9].

### 3.4. Cleavage of S into S2″ by Metalloproteases Occurs Close to the Transmembrane Domain

In the presence of ACE2, the shedding of S by cleavage at an S2″ site (Figure 4A) by one or more endogenous metalloprotease(s) and its enhancement by SKI-1 were next investigated in more detail. Thus, based on the apparent molecular size of S2″ and the known specificity of the metalloproteases ADAM10,17 for processing N-terminal to aliphatic residues such as Leu, Ile, Val [51,52], we generated strategic S Ala-mutants such as KI (K1181A + I1183A), KEI (K1181A + E1182A + I1183A), and KIR (K1181A + I1183A + R1185A) that opposite to our predictions, invariably and significantly enhanced the generation of S2″ accompanied by the loss of S2′ (Figure 4B).

The presence of the last Asn_1194_-glycosylation site (Asn-Xaa-Ser/Thr consensus sequence) preceding the TM domain of proS was used as a guide to localize the S2″ cleavage site. Accordingly, endoglycosidase F digestion of cell extracts from HeLa cells expressing proS, did result in a reduced molecular mass of S2″ (Figure 4C), revealing that the shedding of S to generate S2″ must occur C-terminal to Asn_1194_ somewhere in the stretch ESLIDLQELGKYEQYIKWP_1213_ preceding the TM and comprising residues 1195–1213 (Figure 4A). We, thus, generated a series of Glu-mutations (poly E) of Leu, Ile, Tyr, and Lys residues that may represent P1′ sites for ADAM10 or ADAM17 [51,52]. In addition, we generated the deletion mutant proS-Δ1196-1209 lacking residues 1196–1209. Only the poly E substitution mutant abrogated the formation of S2″, and a slight reduction of S2″ was observed with proS-Δ1196-1209, in absence or presence of overexpressed SKI-1 (Figure 4D).

After binding to ACE2 on the target cell, the transmembrane S-protein changes conformation by association between the heptad repeats HR1 and HR2 (Figure 4A) to form a trimeric structure, leading to fusion between the viral and target-cell membranes [53]. Cell-to-cell fusion assays revealed that the KIR mutant, which enhances S2″ formation, as well as the poly E substitution and proS-Δ1196-1209 all resulted in the loss of cell-to-cell fusion activity (Figure 4E). We mutagenized residues likely recognized by ADAMs [51,52] and situated close to the TM, namely I1210E, K1211E, W1212E, and compared them to L1186A for S2″ formation. As opposed to L1186A, the I1210E mutant did not affect S2″ levels (Figure 5A), in agreement with the cell-cell-fusion assay that suggests an inverse correlation between S2″ levels and fusion (Figure 5B). However, both K1211E and W1212E almost completely lost cell-to-cell fusion activity while slightly increased or decreased S2″, respectively (Figure 5B). Thus, all the above mutations induced loss of cell-to-cell fusion while sometimes modifying the generation of S2″, indicating they most likely affect the HR1-HR2 dimerization of S [54,55], as also observed in HR1 and HR2 mutants within the KIR sequence [56]. Interestingly, we observed that the KIR mutant that results in the loss of cell-to-cell fusion activity in transfected HeLa cells (Figure 4E), also abrogated cellular entry of the S-KIR mutant pseudoparticles into Calu-3 cells (Figure 5C), suggesting that oligomerization of activated S-glycoprotein is needed for entry in HeLa and in Calu-3 cells (Figure 1A).

Since ADAM10 and ADAM17 were reported to cleave S-glycoprotein to generate S2′ in vitro [21], we co-expressed these metalloproteases with proS in HeLa cells (Figure 6A). Our cellular data also showed that overexpression of both ADAM10 and ADAM17 results in an enhanced formation of S2′ (from 1% to 3%) and S2″ (from 30% up to 41% of total S-protein) (Figure 6A), in agreement with the inhibition of S2″ formation by metalloprotease inhibitors (Figure 2C), including ZLDI-8, a relatively specific ADAM17 inhibitor [25]. Under such overexpression conditions and in presence of ACE2, ADAM10, and especially ADAM17, significantly decreased cell-to-cell fusion (Figure 6B), possibly correlating with enhanced S2″ formation (Figure 6A). In contrast, under endogenous metalloprotease conditions, the general metalloprotease inhibitor BB94 drastically reduced S2″ (Figure 2C) but also cell-to-cell fusion (Figure 2D), without affecting the likely furin-generated S2′ (Figure 2C). Thus, the negative effect of S2″ on cell-to-cell fusion becomes significant only under high metalloprotease levels, whereas under their endogenous expression metalloproteases mostly activate fusion.

### 3.5. PCSK9 Enhances the Degradation of ACE2 and Reduces Cell-to-Cell Fusion

SARS-CoV-2 infected patients pretreated with the cholesterol-reducing “statins” exhibited reduced mortality and lower risk of COVID-19 hospitalization [36,37,38]. While directly reducing LDLc levels, and inflammation, statins also enhance the expression [57] and circulating levels of PCSK9 [58]. Overexpression of PCSK9 could influence the Spike glycoprotein levels of SARS-CoV-2 or its receptor ACE2. Our data revealed that incubation of HeLa cells expressing ACE2 and S-glycoprotein with purified recombinant WT PCSK9 did not affect the processing of the S-glycoprotein or the levels of the products S2 and S2′ and S2″ (Figure 7A).

The primarily liver-derived proprotein convertase PCSK9 [32] is a major regulator of LDLc [35] as it enhances LDLR degradation in endosomes/lysosomes, and inhibitors of PCSK9 are now prescribed worldwide for the reduction of LDLc [33,34]. This degradation requires the binding of the catalytic subunit of PCSK9 to the EGF-A domain of the LDLR [59], but also the participation of the C-terminal Cys-His-Rich-Domain (CHRD) of PCSK9 that is critical for the trafficking of PCSK9-LDLR complex to lysosomes [60] (Figure 7B). Of the CHRD’s three modules (M1, M2, and M3) [61], only the M2 module is important for the extracellular activity of PCSK9 on cell surface LDLR [62]. The M2 module of PCSK9 was recently shown to bind an exposed R-X-E motif in some MHC-I receptors, leading to their enhanced degradation in lysosomes, independently from the LDLR [33,63]. We have also recently reported that MHC-I receptors such as HLA-C enhance the PCSK9-induced degradation of the LDLR by efficiently targeting the LDLR-PCSK9-HLA-C complex to lysosomes [64,65].

While recombinant PCSK9 does not enhance the degradation of either proS or its products in HeLa cells expressing ACE2 (Figure 7A), WT PCSK9 (Figure 7B) obtained from the media of HEK293 cells (Figure 7C) [66,67] significantly reduced the levels of ACE2 by ~30% (Figure 7D). This revealed that ACE2 can be targeted by extracellular mature PCSK9 to degradation, similar to LDLR, MHC-I, and other receptors [33]. We next tested the ability of extracellular PCSK9 lacking the CHRD (ΔCHRD; aa 31–152 non-covalently bound to aa 153–455) or the M2 module (ΔM2) (Figure 7B–D) to enhance the degradation of ACE2, as these deletants lose their activity on the LDLR [62] and MHC-I receptors [33,63]. Amazingly, these two deletion mutants further enhanced the degradation of ACE2 (~50%) (Figure 7D). This suggests that different from the LDLR and MHC-I, the routing of the PCSK9-ACE2 complex to degradation compartments does not require the CHRD. Thus, we hypothesized that the catalytic subunit (aa 153–421) and/or prodomain (31–152) of mature PCSK9 form a stable complex with ACE2 [32,61]. These data were corroborated in a functional assay, whereby cell-to-cell fusion was significantly reduced by WT PCSK9, as well as by its ΔCHRD and ΔM2 deletants (Figure 7E). Since the PCSK9-D374Y is a powerful catalytic subunit gain-of-function natural variant that greatly enhances the activity of PCSK9 on the LDLR [61,68], we tested its activity and found that it was also the most active form of PCSK9 on both LDLR and ACE2, as it could reduce by ~60% the levels of LDLR and by ~90% those of ACE2 (Figure 7B,D). Notably, overexpression of ACE2 prevented the degradation of the LDLR by WT PCSK9 (Figure 7D), suggesting that ACE2 may bind better PCSK9 than the LDLR and competes with it for PCSK9 binding. We concluded that the catalytic subunit and/or prodomain of PCSK9 may bind ACE2, similar to the LDLR [69], and send the complex PCSK9-ACE2 to degradation.

**Figure 7 viruses-15-00360-f007:**
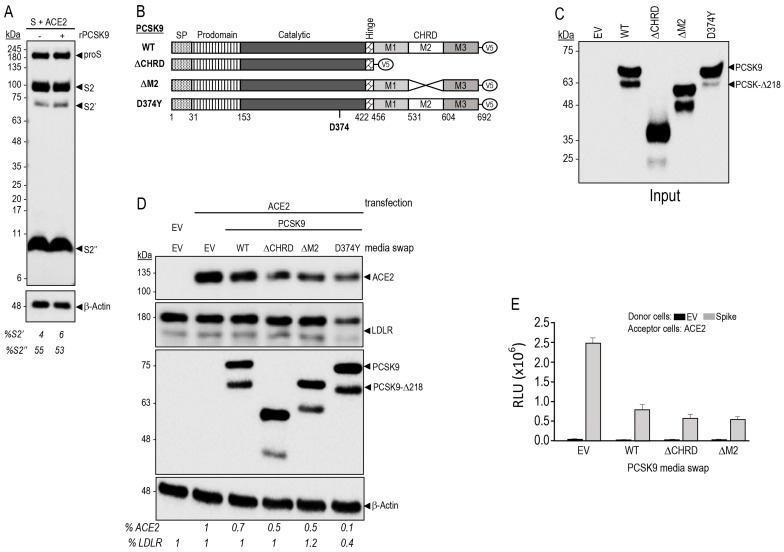
PCSK9 enhances the degradation of ACE2 and reduces cell-to-cell fusion. (**A**) Hela cells transiently transfected with cDNAs encoding WT spike glycoprotein and ACE2 were incubated for 18 h with 5 μg/mL of recombinant PCSK9 [66]. Forty-eight hours post-transfection, proteins in cell lysates were resolved by SDS-PAGE and analyzed by Western blot using mAb-V5. The migration positions of reference molecular weight proteins are indicated. Note that we had to overexpose the gel to better see S2′. (**B**) Schematic representation of V5-tagged WT human PCSK9, PCSK9 domain deletion mutants (ΔCHRD and ΔM2, lacking the CHRD and M2 domains, respectively) and the gain-of-function mutant D374Y. SP: Signal Peptide. (**C**) Conditioned media were produced by transfection of HEK293 cells with cDNAs coding for PCSK9 WT or its mutants. The media were collected 48 h post-transfection, centrifuged 20 min (300× *g* at 4 °C), the production of the conditioned media was then analyzed by Western blot (input). (**D**) HeLa cells were pre-incubated in serum-free-medium for 1 h, followed by the indicated conditioned media swap for 18 h. The cells were then collected and analyzed by Western blot. The levels of ACE2 and LDLR are shown as a percentage of total protein relative to β-actin. (**E**) Donor HeLa cells were transiently transfected with vector expressing empty vector (EV) or spike-glycoprotein WT (S). The acceptor TZM-bl cells were transfected with vector coding for ACE2 receptor. Forty-eight hours post-transfection cells were detached and re-suspended in indicated media (swap) and put in co-culture for 18 h. The extent of fusion is represented as relative luminescence units (RLU).

Since PCSK9 enhances the degradation of both the LDLR and ACE2, we tested the possible implication of LDLR in cell-to-cell fusion of donor Hela cells expressing S and acceptor cells expressing LDLR in absence and presence of ACE2 (Appendix A). Unexpectedly, LDLR alone can serve as a weak receptor for S-glycoprotein, as compared to ACE2 (Appendix A), and furin-cleavage at S1/S2 (Figure 2A) is also required for the LDLR-mediated fusion. Indeed, the furin-resistant mutants μS1/S2 and μAS1/S2 [9] no longer permit LDLR-enhanced fusion (Appendix A). However, expression of ACE2 or LDLR in acceptor cells revealed that ACE2 is a better fusion-inducing receptor for S than the LDLR, and that their co-expression did not enhance the cell-to-cell fusion afforded by ACE2 alone, nor the ability of PCSK9 to reduce cell-to-cell fusion (Appendix A).

The S-glycoprotein lacks an R-X-E motif that may interact with PCSK9, but two such motifs were found in ACE2, namely RSE_171_ at the N-terminal domain and a RSE_699_ close to the TMD (aa 741–765). Molecular modeling of a potential complex of WT PCSK9 with ACE2 (Figure 8A), eliminated the implication of RSE_171_ in ACE2, but suggested that the PCSK9 prodomain R_104_ and R_105_ may form a salt bridge with E_699_ and E_701_ of ACE2 (Figure 8B), close to the RSE_699_ motif in the neck domain (Figure 8B), as well a K_110_ of the prodomain with D_693_ of ACE2. In addition, K_689_ of ACE2 may bind E_210_ in the catalytic domain of PCSK9 (Figure 8A), just before the disordered exposed loop (aa 213–218) [61] containing the furin cleavage site at R_218_ [70] (Figure 8B). Interestingly, the PCSK9′s D_374_ is potentially repulsed by E_56_ in ACE2 (Figure 8C), likely rationalizing the higher affinity of PCSK9-D374Y to ACE2.

## 4. Discussion

Viral envelope glycoproteins directly interact with host cell receptors and mediate virus-host membrane fusion [5], and, therefore, represent key factors in determining host cell tropism. SARS-CoV-2 infection reshapes cholesterol metabolism via gene activation and increased host metabolism activity [71], and depends on the presence of cholesterol-rich lipid raft and endosomal acidification [72]. However, the respective possible implication of the two proprotein convertases SKI-1/S1P and PCSK9, which regulate lipid metabolism and LDL-cholesterol levels [34] in SARS-CoV-2 infection has not yet been addressed. Furthermore, the correlation between LDL-cholesterol levels and COVID-19 disease severity is still unclear.

The ubiquitously expressed SKI-1 is highly conserved among mammalian hosts. SKI-1 plays a critical role in the proteolytic activation of sterol regulatory element binding proteins (SREBPs), which are secretory transcription factors that control the expression of key enzymes involved in cholesterol/fatty acid biosynthesis [28,73]. In addition, SKI-1 was implicated in the direct cleavage/activation of various viral surface glycoproteins including those of arenaviruses, flaviviruses, and hantaviruses [5,74]. SKI-1 activity also indirectly enhances viral infections of several flaviviruses, including hepatitis C [75], dengue [76], zika [77], and hanta [78] viruses via SREBP-induced increased synthesis of cholesterol and fatty acids needed for viral production and/or membrane fusion.

PCSK9 is mostly expressed in hepatocytes, and to a lesser extent in small intestine and kidney [32], and abundantly circulates in plasma [33]. The circulating mature prodomain-PCSK9 non-covalent complex was shown to bind in a non-enzymatic fashion various receptors and escort them to endosomes/lysosomes for degradation [33]. So far, the major reported receptors targeted by circulating PCSK9 are LDLR, VLDLR, ApoER2, LRP1, CD36, and MHC-I [33,64]. The ability of mature PCSK9 to escort the LDLR to lysosomal degradation compartments, led to the development of powerful anti-PCSK9 therapies to reduce the levels of LDLc [79,80], some of which, such as subcutaneous injections of PCSK9 mAb or siRNA, are now prescribed in clinics worldwide [33]. Additionally, the PCSK9-induced degradation of MHC-I receptors paved the way to the use of PCSK9 inhibitors as adjuvants to immunotherapies targeting various cancers [81]. The role of PCSK9 in viral infections is just emerging [33], as exemplified by upregulated expression of PCSK9 during dengue virus infection leading to a reduced innate immune response to the virus [82].

In the present study, we investigated the possible implication of SKI-1 activity in the regulation of cell-to-cell fusion induced by the S-protein upon binding to the receptor ACE2 in acceptor human HeLa cells (Figure 1B), and of the entry of a pseudovirus expressing the spike glycoprotein into the human lung-derived Calu-3 cells endogenously expressing ACE2, furin and TMPRSS2, which occurs primarily by a process at the plasma membrane (Figure 1A) [9]. The data revealed that SKI-1 activity is indirect. SKI-1 enhances cell-to-cell fusion without modifying the processing of S (Figure 1 and Figure 2). Indeed, we showed that SREBP-2 activation is implicated in this process since nSREBP-2 mimics SKI-1 activity and resulted in enhanced cell-to-cell fusion (Figure 3A), likely due to increased cholesterol synthesis, a process also observed in presence of excess mevalonate (Figure 3B), an intermediate in the cholesterol synthesis pathway [34,48]. However, we showed that SKI-1 activity (Figure 1F), nSREBP-2 or mevalonate did not significantly enhance entry of the S-pseudovirus into Calu-3 cells (Figure 3C), possibly due to the presence of endogenous TMPRSS2 (absent in HeLa cells) that exhibits a dominant critical effect on S2′ generation and viral entry [9].

Interestingly, SKI-1 activity also enhanced the generation of S2″ independently from furin (Figure 2). Cleavage at S2″ induces shedding of the membrane-bound S-protein into a soluble and secreted S1_VL_ (Figure 2B). Overexpression of metalloproteases suggested their implication in S2″ cleavage (Figure 2C). Our assumption was that the cleavage of S into S2″ should reduce fusion activity. Consistently, the increased S2″ formation observed with the point mutant L1186A and the S-KIR mutant (Figure 4 and Figure 5) did show a reduced cell-to-cell fusion.

Under endogenous levels of metalloproteases and SKI-1, their respective inhibitors, BB94 and PF-429242, significantly reduced cell-to-cell fusion. The combination of BB94 or PF-429242 with a furin inhibitor (BOS-981) [9] showed an impressive additive inhibitory effect, especially for BOS-981 + PF-429242 (Figure 2D). We concluded that under endogenous conditions the activity of furin, metalloproteases, and SKI-1 together enhance cell-to-cell fusion, which might produce a level of S2″ that is below the threshold to become inhibitory.

Various attempts to identify the S2″ shedding site by mutagenesis and deletions in the S-sequence suggested that S2″ cleavage occur at or close to Lys-Trp_1212_ just before the TM (Figure 4 and Figure 5). While maximal prevention of S2″ formation seen with the W1212E mutant was associated with reduced, rather than increased, cell-to-cell fusion (Figure 5), enhanced formation of S2″ as seen with the single point L1186A mutant (Figure 5) and the S-KIR mutant (Figure 4) did result in reduced cell-to-cell fusion. We concluded that the mutants used, while affecting S2″ levels, might also reduce the propensity of S2 to trimerize, a property dependent on the HR1/HR2 oligomerization domains and the stalk region before the TM [83,84], thus leading to overall decreased cell-to-cell fusion. Notably, none of the identified variants of SARS-CoV-2 exhibit mutations around the S2″ shedding site. In the native S-protein, we observed a significant but partial (~25%) reduction of cell-to-cell fusion upon overexpression of ADAM10 and especially ADAM17 in acceptor HeLa-ACE2 cells undermining the functional importance of increased S2″ formation as a deterrent to cell-to-cell fusion (Figure 6). We concluded that the generation of S2″ by overexpressed ADAMs, which are activated by cholesterol in lipid rafts [23,24], and in vivo under high cholesterol or mevalonate levels [25], is likely to exert a relatively minor effect on cell-to-cell fusion, due to the dominant pro-fusion effect induced by nSREBP-2 increased cholesterol favoring cleavage at S1/S2 [31] and the generation of fusion competent S2′ by furin and metalloproteases in HeLa cells [9], and possibly TMPRSS2 in human airway epithelial cells in vivo [85].

We next turned our attention to the role of PCSK9 in SARS-CoV-2 fusion. Incubation of HeLa cells expressing proS and ACE2 with purified recombinant PCSK9 revealed that PCSK9 does not affect the endogenous processing of proS into S2, S2′ or S2″ (Figure 7A). However, extracellular PCSK9, and especially its gain-of-function D374Y, significantly enhanced the degradation of ACE2 (Figure 7D). The CHRD domain of PCSK9, especially its M2 module, are critical for the trafficking of the PCSK9-LDLR complex to endosome/lysosomes for degradation [33,62]. Amazingly, the critical domains for the PCSK9 activity on ACE2 seem to be the catalytic and/or prodomain of PCSK9 and not the CHRD, making ACE2 a unique and novel PCSK9-target quite distinct from the LDLR and MHC-I [33]. Molecular modeling of the possible PCSK9-ACE2 complex suggested that the PCSK9 prodomain R_104,105_ may form a salt bridge with E_699,701_ of ACE2 (Figure 8B) situated within the segment needed for the shedding of ACE2 by TMPRSS2 [9], and that the catalytic subunit of the PCSK9 gain-of-function D374Y is better poised to interact with ACE2. Indeed, the E_56_ in ACE2 is predicted to no longer be repulsed by D_374_ found in WT PCSK9, as it would be replaced by Y_374_ in PCSK9-D374Y (Figure 8C). This is in contrast to enhanced binding of PCSK9-D374Y to LDLR due to a salt bridge between the PCSK9′s Y_374_ and H_306_ in the EGF-A domain of the LDLR [86]. Interestingly, the R_104_ has been shown to exert an important role in PCSK9′s function on the LDLR since R104C [87], R104Q [88], and R104H (Seidah, unpublished) result in a loss-of-function. This may in part rationalize why ACE2 seems to inhibit PCSK9′s ability to enhance the degradation of the LDLR (Figure 7D). Recently, we reported a detailed analysis of the PCSK9-LDLR complex trafficking to endosomes/lysosomes for degradation [64,65]. The study implicated two other proteins for the effective targeting of the complex for lysosomal degradation: the cyclase associated protein 1 (CAP1) binding the prodomain and the M1/M3 domains of PCSK9 and HLA-C (an MHC-I family member) binding the M2 domain of PCSK9. The sorting signal seems to be present within the cytosolic tail of HLA-C, as that of the LDLR is not effective to drag the complex PCSK9-LDLR to lysosomes [89]. Analysis of the sequence of the cytosolic tail of ACE2 reveals the presence of a C-terminal Thr-X-Phe_805_ motif, which fits the PDZ-binding motif S/T-X-Ø (where X denotes any residue, and Ø denotes a hydrophobic residue) [90]. Whether this motif regulates the trafficking of the PCSK9-ACE2 complex to reach lysosomes for degradation versus cell surface recycling [91] is still unknown.

While the proposed PCSK9-ACE2 interaction model is still speculative, it opens new avenues in our understanding of the novel PCSK9-ACE2 interaction, which may act as a countermeasure and limit the complications of ACE2-dependant SARS-CoV-2 infection. Whether the clinically prescribed PCSK9 mAb (Evolocumab-Repatha and/or Alirocumab-Praluent) that target the catalytic subunit [33] would also prevent the function of PCSK9 on ACE2 is yet to be defined, as well as the potential effect of these LDLc-reducing treatments on SARS-CoV-2 infection and/or spread. In that context, very recently it was reported that hospitalized COVID-19 patients receiving a single subcutaneous injection of the PCSK9-mAb evolocumab exhibited reduced death or need for intubation, as well as decreased inflammatory cytokine IL-6 levels in severe COVID-19 cases [92]. Our data in the present study and the outcomes of the above clinical trial suggest that lack of active circulating PCSK9 leading to reduced inflammation, as was reported before [33,93,94,95,96], may in part be due to enhanced levels of ACE2 activity that would generate higher levels of the anti-inflammatory angiotensin 1-7 [97,98]. Indeed, the potential of angiotensin 1-7 in the treatment of critically ill COVID-19 patients is now being evaluated in ongoing clinical trials [98]. Finally, it was reported that COVID-19 patients who are taking “statins” exhibit reduced mortality and risk of hospitalization [36,37,38]. Since statins reduce inflammation but also increase the transcription of the *PCSK9* gene [57] as well as the circulating levels of the protein [58], it would be informative to measure plasma PCSK9 levels in COVID-19 patients taking or not taking statins and correlate them with disease severity and hospitalization status.

As illustrated by the graphical model in Figure 9, the data presented in this work revealed that the cholesterol-regulating SKI-1 and PCSK9 can modulate the cell-to-cell fusion induced by SARS-CoV-2 spike interaction with ACE2. SKI-1 enhanced cell-to-cell fusion via increased SREBP-2 activity and metalloprotease activation and conversely PCSK9 reduced cell-to-cell fusion by enhancing the degradation of ACE2 in a novel pathway requiring the prodomain/catalytic subunit of mature PCSK9.

Finally, the cellular infection of SARS-CoV-2 while dependent on TMPRSS2 and furin activity, is quite different from that of SARS-CoV-2 bearing Omicron variants, as the viral entry of the latter seems to be independent of TMPRSS2 activity and/or levels [99,100,101]. It would be informative if such Omicron variants that could use the endocytosis pathway (Figure 1A) may also depend on SKI-1 activity for viral entry and/or replication, like other RNA viruses [5].

**Figure 9 viruses-15-00360-f009:**
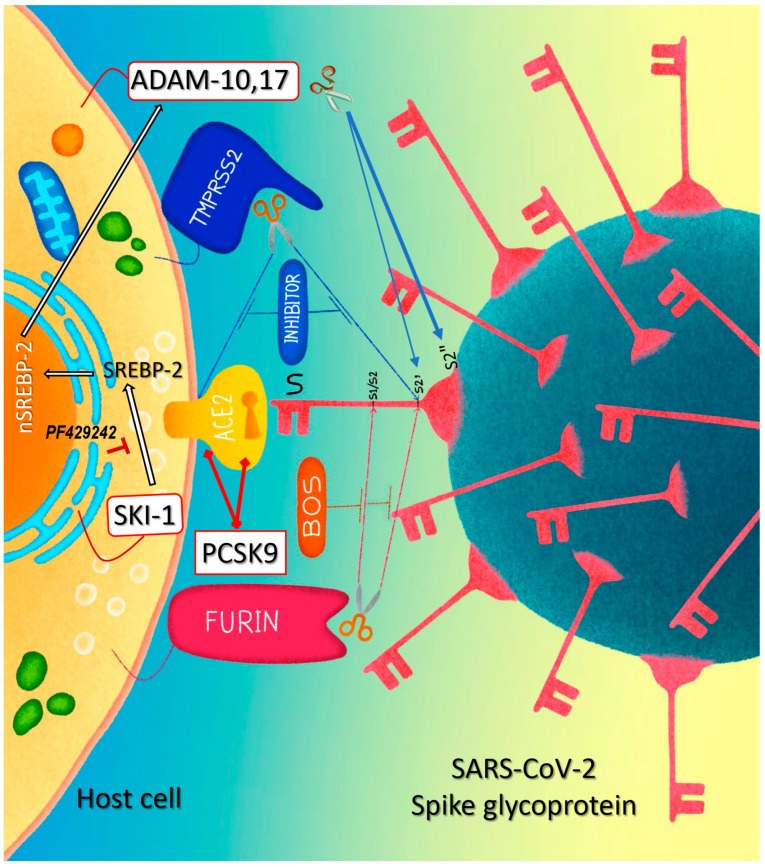
Graphical representation of a working model of SKI-1 and PCSK9 effects on cell-to-cell fusion. At low level of sterols in the cell, SREBP-2 moves from the ER to Golgi where it is cleaved by SKI-1 and then activated by S2P [29], the N-terminal domain of SREBP-2 (nSREBP-2) is then translocated to the nucleus and triggers the transcription of genes that are involved in cholesterol synthesis [102]. In turn, cholesterol increases the activity of metalloproteinases at cell surface, notably ADAM10 and ADAM17 [24]. The activated metalloproteinases enhance the cleavage of the S-glycoprotein at S2′ and increase cell-to-cell fusion [21]; they also shed the protein by cleaving it at S2″ close to the TMD (this work). The cleavage at S2′ site can also be achieved by furin and TMPRSS2, and the latter sheds ACE2 as previously shown [9]. The combination of SKI-1 and/or metalloproteinase inhibitors with a furin-like inhibitor can be a potent tool to control cell-to-cell fusion and viral entry. Extracellular PCSK9 enhances the degradation of ACE2 by a mechanism requiring only the prodomain/catalytic subunit of mature PCSK9 that likely bind ACE2, thereby reducing cell surface levels of ACE2, cell-to-cell fusion, and possibly SARS-CoV-2 infection.

## Figures and Tables

**Figure 1 viruses-15-00360-f001:**
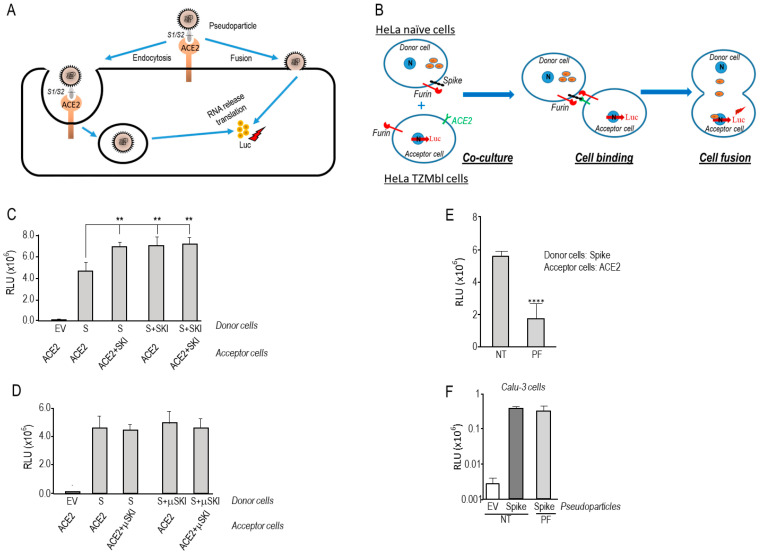
SKI-1 activity is required for cell-to-cell fusion in HeLa cells but not for pseudoparticle entry. (**A**) Schematic representation of pseudotyped particles’ entry into Calu-3 cells. Upon incubation of cells with nanoluciferase-expressing HIV particles pseudotyped with SARS-CoV-2 S WT, pseudoparticles can either enter the cell by endocytosis or fusion upon Spike priming and activation. After RNA release and nanoluciferase expression, the extent of fusion is quantified by measuring nanoluciferase activity. (**B**) Cell-to-cell fusion between donor cells (HeLa) expressing the SARS-CoV-2 spike protein along with the HIV trans-activator Tat, and acceptor cells (TZM-bl) that express ACE2. Upon fusion, Tat is transferred from donor to acceptor cells, thereby inducing luciferase expression. (**C**,**D**) Hela cells transiently transfected with an empty vector (EV) or expressing SARS-CoV-2 spike (S) (donor cells), were co-cultured for 18 h with TZM-bl HeLa cells expressing ACE2 receptor (acceptor cells). The values shown are the (**C**) WT SKI-1 or (**D**) its active site mutant (μSKI-1, H249A), which were expressed in acceptor and/or donor cells. (**E**) Donor HeLa cells expressing WT-S were co-cultured with acceptor TZM-bl cells expressing ACE2 in absence or presence of 10 μM of SKI-1 Inhibitor (PF429242). (**F**) Calu-3 cells were inoculated with nanoluciferase-expressing HIV particles pseudotyped with empty vector (EV) or SARS-CoV-2 wild-type spike (WT), in the absence or the presence of 10 μM of SKI-1 Inhibitor (PF429242). The cell-to-cell fusion luciferase assay used throughout is detailed in the Material and Methods Section 2.4. The relative light units (RLU) were obtained using a Promega GloMax plate reader luminescence detection system. Representative data from at least three independent experiments are shown. *p* values (**, *p* < 0.01; ****, *p* < 0.0001) were evaluated by a Student’s *t*-test.

**Figure 2 viruses-15-00360-f002:**
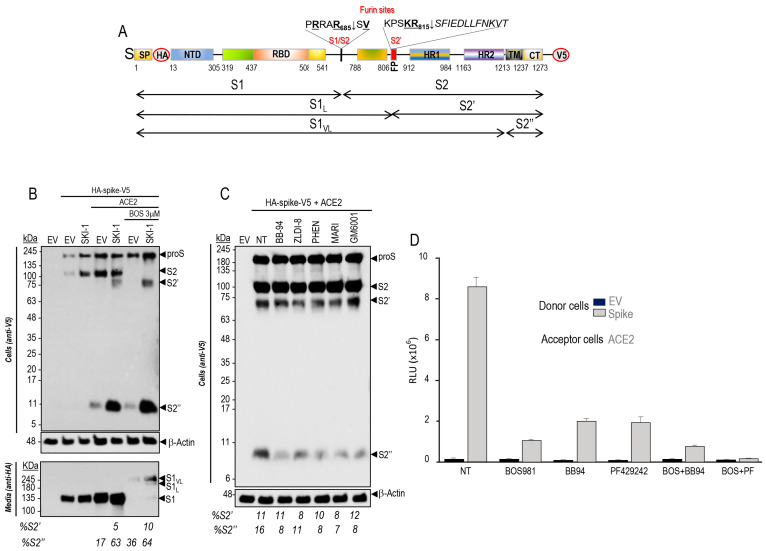
SKI-1 increases the cleavage at S2′ and sheds the spike glycoprotein. (**A**) Schematic representation of the primary structure of preproS, including its domains, the predicted furin-like S1/S2 site generating the S1- and S2-subunits, and the S2′ site preceding the fusion peptide (FP). The signal peptide (SP), N-terminal domain (NTD), receptor binding domain (RBD) to ACE2, the two heptad repeats HR1 and HR2, the transmembrane domain (TM), the cytosolic tail (CT) and the C-terminal V5-tag are indicated. (**B**) Western blot analyses of cell extracts and media from HeLa cells following co-transfection of cDNAs coding for doubly tagged WT proS with empty vector (EV) and/or that coding for ACE2 and SKI-1, and treatment with vehicle control (DMSO) or with 3 μM of the furin inhibitor BOS-981. The cell extracts (upper panel) and the media (lower panel) were analyzed using anti-V5 and anti-HA mAbs, respectively. The sizes and schematics of the generated fragments are indicated in (**A**), and the migration positions in (**B**). (**C**) Hela cells transiently transfected with a cDNA encoding an empty vector (EV) or with one expressing the V5-tagged spike (S) glycoprotein were treated, 24 h post- transfection, with vehicle DMSO (NT) or with 1 μM of the indicated metalloproteinase inhibitors. The cell extracts were analyzed by Western blotting using an mAb-V5. The percentage of shed S2″ is indicated. (**D**) Naive Hela cells (donor) were transfected with vectors expressing either no protein (EV) or WT-spike (S). TZM-bl HeLa cells (acceptor) were transfected with a vector expressing ACE2. After 48 h, donor and acceptor cells were co-cultured for 18 h in the absence (NT) or presence of the furin inhibitor (BOS-981, 3 μM) and/or metalloproteinase inhibitor (BB, 1 μM) and SKI-1 inhibitor (PF429242, 10 μM). The extent of fusion is represented as relative luminescence units (RLU).

**Figure 3 viruses-15-00360-f003:**
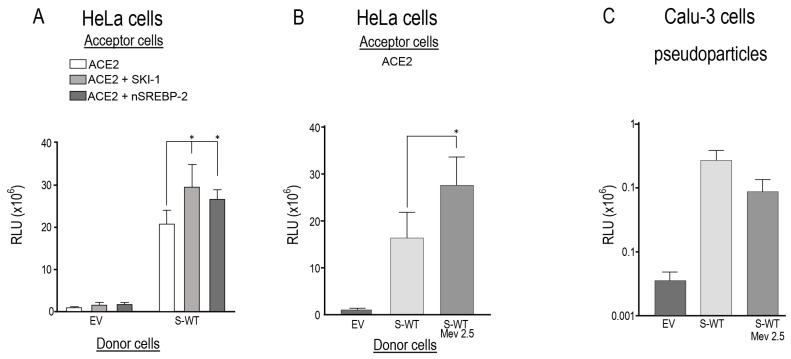
SKI-1 and nSREBP-2 enhance cell-to-cell fusion. (**A**) HeLa cells were transiently transfected with an empty vector (EV) or one encoding WT-spike (S). Acceptor TZM-bl cells were transfected with a vector expressing ACE2 alone or with vector expressing either SKI-1 or nSREBP-2. After 48 h, donor and acceptor cells were co-cultured for 18 h. (**B**) Donor cells were transfected with vectors expressing either no protein (empty vector, EV) or WT-spike (S) were co-cultured for 18 h with acceptor TZMbl HeLa cells expressing ACE2 receptor. Prior to co-culture, the cells were treated with 2.5 mM mevalonate. Relative luminescence units (RLU) were normalized to the EV value arbitrarily set to 1. Data are presented as mean values ± SD (n = 3 independent experiments). (**C**) Calu-3 cells were transduced with nanoluciferase-expressing HIV particles pseudotyped with empty vector (EV) or SARS-CoV-2 wild-type spike (WT), in absence or presence of 2.5 mM of Mevalonate (Mev). Pseudoparticle entry was expressed as relative luminescence units (RLU). Representative blots of at least three independent experiments are shown. *p* values (*, *p* < 0.05) were evaluated by a Student’s *t*-test.

**Figure 4 viruses-15-00360-f004:**
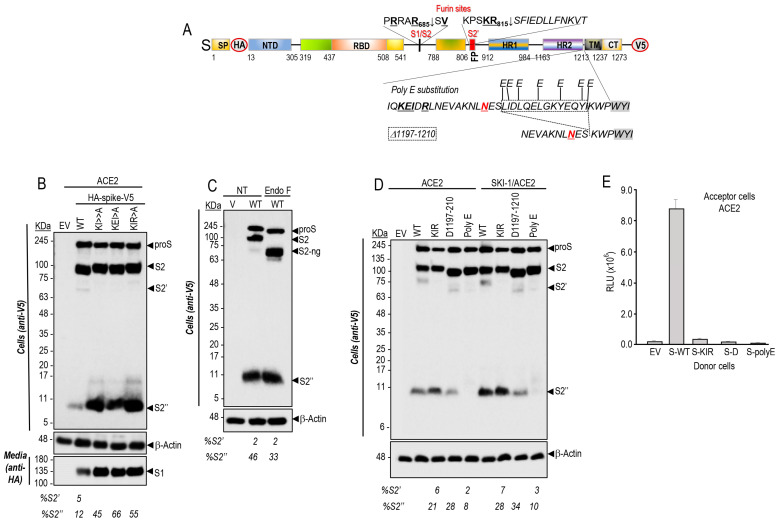
Identification of a potential shedding site of spike glycoprotein. (**A**) Schematic representation of the primary structure of preproS showing WT sequence and the mutants of the C-terminal of HR2 domain. (**B**) Western blot analyses of HeLa cells following co-transfection with cDNAs coding for ACE2 and either WT-S-protein or its mutants: double KI (K1181A + I1183A) or triples KEI (K1181A + E1182A + I1183A), and KIR (K1181A + I1183A + R1185A) mutants. Note that mutants significantly enhanced the generation of S2″ accompanied by the loss of S2′. (**C**) Extracts from HeLa cells transfected with empty vector (EV) or V5-tagged wild type spike-protein (WT) were treated with Endo-F or mock treated (NT) and analyzed by Western blot using anti-V5 antibody. Note the molecular shift of non-N-glycosylated forms (ng) proS, S2 and S2′ but not that of S2″ fragment after endo-F treatment. (**D**) Western blot analysis using mAb-V5 of cell lysates from HeLa cells expressing ACE2 with empty vector (EV), wild type spike-protein (WT), or its mutants: KIR (K1181A + I1183A + R1185A), Δ-S (Δ1197-1210), or poly E substitution in the absence or presence of SKI-1. (**E**) Cell-to-cell fusion assay of donor HeLa cells expressing either empty vector (EV), wild type spike-protein (WT), or its mutants: KIR, Δ-S, or poly E substitution with acceptor cells expressing ACE2 receptor. All the mutants in the HR2 domain abrogate cell-to-cell fusion.

**Figure 5 viruses-15-00360-f005:**
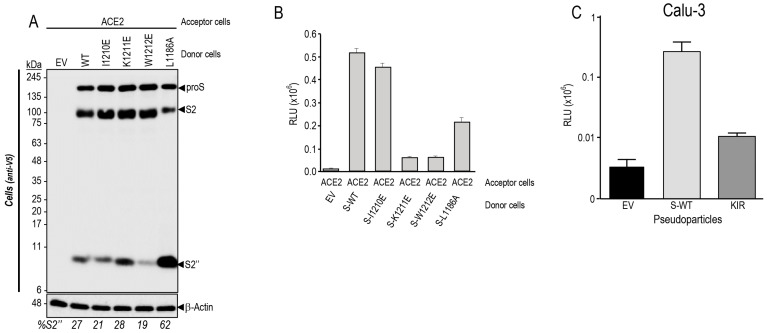
Mutations of proS in HR1 domain prevent cell-to-cell fusion. (**A**,**B**) Donor Hela cells expressing either no protein (EV), WT-spike (S), or its indicated mutants were co-cultured for 18 h with acceptor TZM-bl HeLa cells transfected with a vector expressing ACE2 receptor. (**A**) Western blot analysis of spike processing in cell extracts. (**B**) Cell-to-cell fusion: the extent of fusion is represented as relative luminescence units (RLU). (**C**) Impact of KIR mutation on pseudoparticle entry in Calu-3 cells.

**Figure 6 viruses-15-00360-f006:**
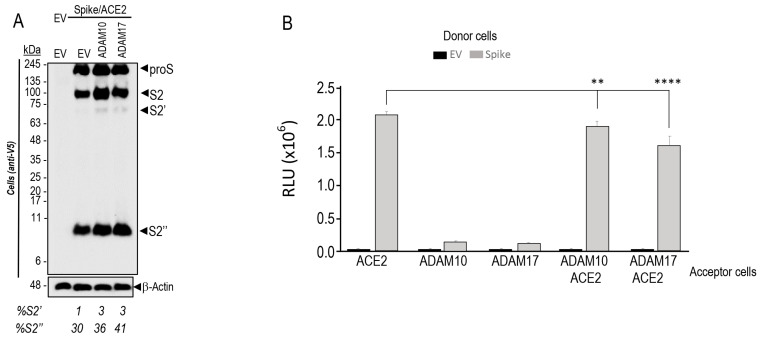
Overexpression of ADAM10 and ADAM17 increases the generation of S2′ and S2″ products but decreases cell-to-cell fusion. (**A**) HeLa cells were transiently co-expressed with double-tagged spike protein (N-terminal HA-tag; C-terminal V5-tag), WT (S), and ACE2 alone or in combination with empty vector (EV), ADAM10 or ADAM17. The V5 immunoblot shows the analysis of cell extracts 28 h post transfection. Note that we had to overexpose the gel to better see S2′. (**B**) Donor HeLa cells expressing empty vector (EV) or spike-glycoprotein WT (S) were co-cultured with acceptor TZM-bl cells expressing ACE2 and/or ADAM10 and ADAM17. The extent of fusion is represented as relative luminescence units (RLU). Representative blots of at least three independent experiments are shown. *p* values (**, *p* < 0.01; ****, *p* < 0.0001) were evaluated by a Student’s *t*-test.

**Figure 8 viruses-15-00360-f008:**
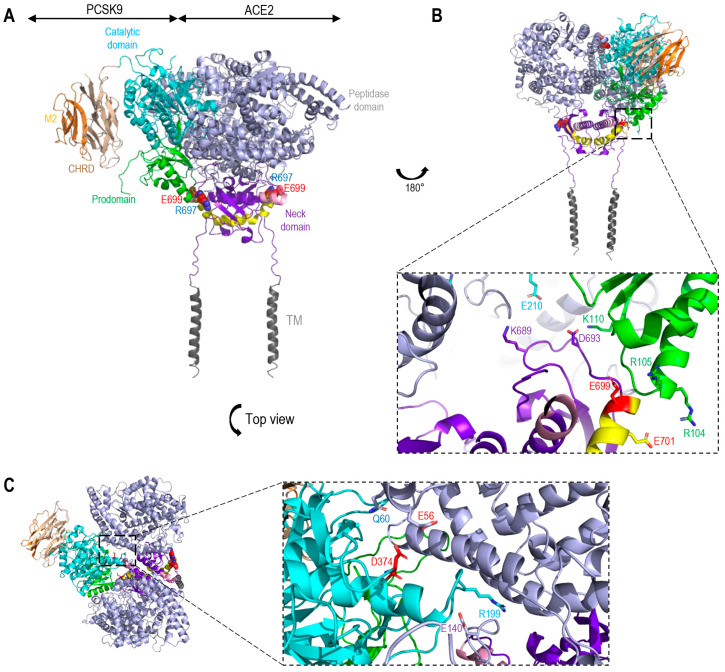
Molecular modeling of the interaction of OCSK9 and ACE2. (**A**) The catalytic domain (cyan) and prodomain (green) of PCSK9 (PDB code:2P4E) interacts with the extracellular domain of ACE2 (PDB code:6m18) (grey and purple). Cleavage site ADAM17 (pink). Cleavage site TMPRSS2 (yellow). (**B**) Rotation 180°, view on the binding site between the prodomain of PCSK9 (R_104_ and R_105_) and shedding domain by TMPRSS2 in ACE2 (E_699_ and E_701_) [9]. Additional binding between K_110_ of the prodomain and E_210_ of the catalytic domain of PCSK9 with D_693_ and K_689_ of the Neck domain of ACE2. (**C**) Blown up top view of the binding site between Catalytic domain of PSK9 (D_374_ and R_199_) and peptidase domain of ACE2 (Q_60_ and E_140_). Notice the predicted repulsion between D374 in PCSK9 and E56 in ACE2 (in red). TM, transmembrane domain.

## Data Availability

The authors confirm that the data supporting the findings of this study are available within the article and/or its Appendix A. Source data are provided with this paper. The data that support the findings of this study are preserved at repositories of the Montreal Clinical Research Institute (IRCM), Montreal, QC, Canada, and available from the corresponding authors upon reasonable request.

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
