# Peer review of "SKI-1/S1P Facilitates SARS-CoV-2 Spike Induced Cell-to-Cell Fusion via Activation of SREBP-2 and Metalloproteases, Whereas PCSK9 Enhances the Degradation of ACE2"

_viruses, 2023, doi:10.3390/v15020360_

Round 1

Reviewer 1 Report

Comments

This team has published several papers regarding biochemical concerns of PCSK9 and SKI-1/S1P implicated in infectious disease.  In this report, the authors tried to elucidate that SKI-1/S1P makes  SARS-CoV-2 Spike easy to induce cell-to-cell fusion via activation of SREBP-2. And they suggest that a couple of metalloproteases and PCSK9 may be potential therapeutic targets against SARS-CoV-2 infection. They have insisted that the cholesterol-regulating SKI-1 and PCSK9 modulate the cell-to-cell fusion leading to induce SARS-CoV-2 spike interaction to ACE2. They are showing wide spectrum of biochemical proves in vitro. However, there are many logical gaps in the evidence to prove their claim. To increase the significance of this article, they not only need to provide detailed evidence, but results should be validated in vivo including animal model or clinical test.

Major concerns are;

1)     In Fig 1. Luc activities were increased SKI-1-dependent manner. To explore the potential role of SKI-1 to increase cell-to cell fusion, please precisely explain behind the Fig 1C and 1D. And, considering localization of Golgi-residency of SKI-1, there should be any transcription factor caused by cellular trafficking. It would be much better to prove localization of transfected SKI-1 as well.

2)     In Fig 2D is showing that PF-429242 + BOS-981, an inhibitor for SKI-1 activation, has a strong inhibitory effect on cell-to-cell fusion. The authors failed to include any change of SKI-1 substrates from PF treated group for a better understanding and comparison of fusion assay.

3)     Very hard to understand meaning of the Fig 3. SREBP processing is a negative and stimulus feedback system. Upon cholesterol depletion, the SREBP-Insig-SCAP complex dissociates, and SCAP escorts SREBPs from the ER to the Golgi apparatus, followed by S1P-initiated release of the N-term domain entering into the nucleus. So, cholesterol depletion is a trigger for SREBP-processing, but not mevalonate. Authors need to show the localization of SREBP-2. Otherwise, as mentioned in text, activities of HMGCR, MVK, or LDLR depends on changes of mature form of SREBP-2. Please show their activities with mRNAs or proteins.

4)     One of the critical defacts regrding the Fig 9 graphical model is on SREBP-2. As implied above, the SREBP-2 process in ER to Golgi apparatus in response to cholesterol deprivation. So, relocate SREBP-2 to ER. Do you have any direct evidence that SREBP-2 regulate ADAM-10 or -17 ?  Please show them.

Author Response

Answers to Reviewer number 1

1- Figure 1: Luc activities were increased SKI-1-dependent manner. To explore the potential role of SKI-1 to increase cell-to-cell fusion, please precisely explain behind the Fig 1C and 1D. And considering localization of Golgi-residency of SKI-1, there should be any transcription factor caused by cellular trafficking. It would be much better to prove localization of transfected SKI-1 as well.

For clarification, we have now added to the legend of Fig. 1 the following sentence:

“The cell-to-cell fusion luciferase assay used throughout is detailed in the Material and Methods section 2.4. The relative light units (RLU) were obtained using a Promega GloMax plate reader luminescence detection system.”

We already published the cellular localization of transfected SKI-1 and its active site mutant H249A, whereby SKI-1 was localized in paranuclear structures reminiscent of the cis/medial Golgi whereas its H249A mutant was localized in the endoplasmic reticulum. (PMID: 17623657).

2- Figure 2: In Fig 2D is showing that PF-429242 + BOS-981, an inhibitor for SKI-1 activation, has a strong inhibitory effect on cell-to-cell fusion. The authors failed to include any change of SKI-1 substrates from PF treated group for a better understanding and comparison of fusion assay.

We thank the referee for this suggestion. Accordingly, to define the inhibitory potency and specificity of PF-429242 inhibitor toward SKI-1 activity we selected human proBDNF. The latter is cleaved by SKI-1 into a 28 kDa intermediate which is also processed by Furin into the 12.5 kDa mature BDNF (see PMID: 9990022). Thus, we co-expressed in HeLa cells proBDNF with SKI-1 or Furin in the presence or absence of the PF-429242 SKI-1 inhibitor. The data in the new Supplementary Figure S1 clearly showed that PF-429242 abrogates the SKI-1 ability to process proBDNF into the 28 kDa intermediate but not that of Furin to generate the 12.5 kDa BDNF. We have now added this Figure as a new Supplementary Figure S1.

Accordingly, we added the following text in the results section 3.2:

To validate the functionality of the cell-permeable SKI-1 inhibitor PF-429242 in HeLa cells [43, 44], we selected proBrain Derived Neurotrophic Factor (proBDNF) as a substrate, since it was reported to be first cleaved by SKI-1 into a 28 kDa intermediate, which together with proBDNF are cleaved by furin to produce a 12.5 kDa mature BDNF [26]. Accordingly, co-expression of proBDNF with either SKI-1 or furin generated the expected products, but the incubation of cells with PF-429242 selectively eliminated the formation of the 28 kDa product, without affecting the generation of mature BDNF by furin (Supplementary Figure S1).

3- Very hard to understand meaning of the Fig 3. SREBP processing is a negative and stimulus feedback system. Upon cholesterol depletion, the SREBP-Insig-SCAP complex dissociates, and SCAP escorts SREBPs from the ER to the Golgi apparatus, followed by S1P-initiated release of the N-term domain entering into the nucleus. So, cholesterol depletion is a trigger for SREBP-processing, but not mevalonate. Authors need to show the localization of SREBP-2. Otherwise, as mentioned in text, activities of HMGCR, MVK, or LDLR depends on changes of mature form of SREBP-2. Please show their activities with mRNAs or proteins.

At low levels of sterols in the cell, SREBP-2 moves form the ER to Golgi where it is cleaved by SKI-1 and then activated by S2P, the N-terminal domain of SREBP-2 is then translocated to the nucleus (nSREBP-2) and triggers the transcription of genes that are involved in cholesterol synthesis. Mevalonate is the 3rd intermediate product in the cascade leading to cholesterol synthesis. To bypass the activation of SREBP-2 by SKI-1, we incubated the HeLa cells with 2.5 mM mevalonate or over-expressed the N-terminal soluble domain of SREBP-2 (nSREBP-2). In both cases the cell-to-cell fusion was increased (Fig 3.). This data indicated a role for SKI-1 in the cell-to-cell fusion via enhanced cholesterol synthesis. It is now explicitly indicated in the text and in Figure 3 that we transfected a cDNA encoding nSREBP-2 and not the full length SREBP-2.

4- One of the critical defacts regarding the Fig 9 graphical model is on SREBP-2. As implied above, the SREBP-2 process in ER to Golgi in response to cholesterol deprivation. So, relocate SREBP-2 to ER. Do you have any direct evidence that SREBP-2 regulate ADAM-10 or -17 ?  Please show them.

As requested by the referee we have now explicitly shown in the new Fig. 9 that the SKI-1 cleavage of SREBP-2 occurring the cis/medial Golgi, will ultimately result in the generation of soluble cytosolic nSREBP-2 that is translocated to the nucleus. In turn this would result in enhanced cholesterol synthesis leading to higher levels of active ADAM-10/17.

Indeed, in our recent study demonstrated that, in vivo, the shedding of the LDLR by ADAM10/ADAM17 is cholesterol-dependent and this shedding is enhanced by mevalonate or high cholesterol feeding in mice (PMID:35285474). These data strongly suggest that the SREBP-2 could activate ADAM10/ADAM17 via enhanced cholesterol synthesis.

Accordingly, in the present study we demonstrated that cell-to-cell fusion is directly linked to SKI-1 activity, which ultimately enhances the activation of SREBP-2 into nSREBP-2, resulting in increased cholesterol synthesis. This effect was mimicked by mevalonate treatment or transfection of the active nuclear nSREBP-2 both of which increase cell-to-cell fusion.

Accordingly, we wrote in the discussion:

We concluded that the generation of S2” by overexpressed ADAMs, , which are activated by cholesterol in lipid rafts [23, 84] and in vivo under high cholesterol or mevalonate levels [55], is likely to exert a relatively minor effect on cell-to-cell fusion, due to the dominant pro-fusion effect induced by nSREBP-2 increased cholesterol favoring cleavage at S1/S2 [29] and the generation of fusion competent S2’ by furin and metalloproteases in HeLa cells [9], and possibly TMPRSS2 in human airway epithelial cell in vivo [84].

Reviewer 2 Report

The article focuses on the role of cholesterol-regulating convertases SKI-1 and PCSK9 in the entry and fusion of SARS-CoV-2. The authors try to demonstrate that SKI-1 mediates fusion by enhancing expression of metalloproteases ADAM10 and ADAM17 that can cleave spike at the plasma membrane. In contrast plasma soluble PCSK9 decreases S-mediated fusion by inducing ACE2 degradation. The study focuses on interesting aspects of SARS-CoV-2 biology, but experiments performed by the authors are not sufficient to support their claims. In my view the authors need to address some serious issues before the article is accepted for the publication. In particular I have major concerns about fusion experiments.

1.     The authors claim to perform three independent experiments, but only show one representative experiment. Differences in many experiments are mild but the authors claim they are significant. This is because error bars are from technical replicates (duplicates, triplicates?) and the variability within a single experiment is small. But any statistics including student’s T-test should be done on data combined from at least three independent experiments and not from a single one. As differences between conditions are rather small it is very likely that after combining data from few experiments with bigger variability, the results will not be statistically significant. In my view the authors should show combined data and moderate their language on significancy of differences observed.

2.     Fusion assays are performed in one cell line and with luciferase it is impossible to assess how many cells actually fuse. For this, the authors should include more assays to prove their claims at least for experiments in Figure 1. For that they could use similar assay in other cell line (like A549), use split-GFP system to quantify fluorescence or at least show some phase-contrast images as syncytia are usually clearly visible.

3.     Authors use pseudoparticles for control infections in Calu3 cells only but pseudoparticles entry system could be used to support fusion data in cells like 293T ACE2 or similar. That could definitely support their claims as I find fusion experiments with small luciferase activity differences not convincing enough.

4.     Authors only mention TMPRSS2 when showing no effect in Calu3 cells. It would be greatly valuable to compare SKI1 effect to TMPRSS2 effect. For that the authors should add condition when cells are transfected with ACE2+TMPRSS2. 

5.     In 3.2 the authors conclude that SKI1 increases fusion by acting on metalloprotease activities. However, in 3.3 the authors also investigate the effect of SREBP-2 and the mevalonate pathway and conclude that increase in cholesterol synthesis can also lead to fusion activation. If that is the case the entry of pseudotypes into Calu3 should be affected as well. Therefore, conclusions drawn by the authors are not really supported by the experiments. This should be investigated with some additional experiments or conclusions should be modified.

6.     Experiments with PCSK9 and ACE2 should be repeated in a cell line with endogenous levels of ACE2 (like Calu3). In transfected cells levels of ACE2 are very high and I am not convinced that depletion of 30% shown in Figure 7D is significant. The fact that there is no LDLR depletion with the WT PCSK9 could also suggest that overexpression of ACE2 or LDLR is not a good system to show the real effects of PCSK9. The authors suggest in the discussion that the lack of LDLR degradation could be due to the presence of ACE2 but that should be supported by a control where only LDLR is expressed.

7.     Increase in fusion with LDLR expression is very close to EV control and therefore I don’t think this is significant.

8.     In Figures 2 and 4 on S blots levels of S2” are rather small with ACE2 transfection and they increase with SKI1. But on Figures 6 and 7 it’s very high. How to explain that?

9.     In the discussion the authors widely discuss data they obtained with a lot just being repeated from the results section.  It would be interesting to learn more about implications these results could have for human infection. For example, the omicron variant has been shown to be less reliant on TMPRSS2 than other variants. That usually results in endocytosis but maybe in some cell types SKI1 activation could increase plasma membrane fusion. 

Minor points:

-       In Calu3 cell fusion happens at the plasma membrane due to high levels of TMPRSS2 so labelling Figure 1A as Calu3 cell should be removed.

-       Figure 1S shows furin as a transmembrane protein which is not the case

-       If P values are used it should be shown on graphs which conditions are compared.

-       Figure 2A, furin shown as part of spike? The schematic is misleading.

-       Figure 2D wrong labelling for EV and Spike.

-       Figure 3 data are shown as fold RLU over EV. Why is if different to other figures? 

-       % of S2’ and S2” is shown for some lanes on Western Blots but not for others, why?

-       On page 13, third paragraph, the authors talk about Figure 1A, B where is should be Supplementary Figure 1A, B.

-       In the graphical model furin is again shown as transmembrane protein.

Author Response

Manuscript ID: viruses-2120986                                           

Answers to Reviewer number 2

1- The authors claim to perform three independent experiments, but only show one representative experiment. Differences in many experiments are mild but the authors claim they are significant. This is because error bars are from technical replicates (duplicates, triplicates?) and the variability within a single experiment is small. But any statistics including student’s T-test should be done on data combined from at least three independent experiments and not from a single one. As differences between conditions are rather small it is very likely that after combining data from few experiments with bigger variability, the results will not be statistically significant. In my view the authors should show combined data and moderate their language on significancy of differences observed.

We are agree with the reviewer that the data should be presented as an average of at least three independent experiments, however, in Hela cells, the fusion assay can be affected by many parameters such as cell passage number, cell viability and efficiency of cell transfection, leading to large variations between experiments, so the difference between the control ant treated condition will be not significant in combined independent experiment. Thus, we performed three independent experiments with six replicates to minimize the variations, but with the same trend. An example of the data is presented below. We will replace the three stars by two stars in Figures 1C and 1D.

2- Fusion assays are performed in one cell line and with luciferase it is impossible to assess how many cells actually fuse. For this, the authors should include more assays to prove their claims at least for experiments in Figure 1. For that they could use similar assay in other cell line (like A549), use split-GFP system to quantify fluorescence or at least show some phase-contrast images as syncytia are usually clearly visible.

The number of cells that fuse during cell-to-cell fusion assay was estimated at 50% in our previous study [see reference # 9].

Fusion assay

The cell co-cultures were plated on glass coverslips. After 18-24h, the cells were incubated with 488 CellMask™ to stain the membrane and then fixed with 4% PFA for 15 min at 4ºC. The glass coverslips were mounted on glass slides using ProLong™ Gold Antifade containing DAPI (Invitrogen). The number of syncytia were counted over 10 fields (arrowhead indicates cell fusion).

Cell-to-cell fusion was observed when donor cells are expressing spike WT but not with the empty vector (EV) or Furin cleavage site mutant (Spike mS1/S2). The cell-to-cell fusion was estimated at 50%. (see PMID: 35658532)

We had already indicated in the original text a the beginning of the results section 3.1:

To investigate the potential regulation of SARS-CoV-2 fusion by SKI-1, we performed a cell-to-cell fusion assay based on the co-culture of donor Hela cells transiently expressing wild type (WT)-S and HIV-Tat with acceptor tat-driven luciferase reporter Hela TZ-Mbl cells, transiently expressing ACE2, as described in [9].

3- Authors use pseudoparticles for control infections in Calu3 cells only but pseudoparticles entry system could be used to support fusion data in cells like 293T ACE2 or similar. That could definitely support their claims as I find fusion experiments with small luciferase activity differences not convincing enough.

We agree that using HEK293T over-expressing ACE2 for the pseudoparticles entry will definitively give much higher luciferase activity. We do not have these cells and cannot produce them in the 10-days allotted for this review. However, we did use the available Calu3 cells that express endogenously ACE2 in order to compare them to Hela cells overexpressing ACE2.

4- Authors only mention TMPRSS2 when showing no effect in Calu3 cells. It would be greatly valuable to compare SKI1 effect to TMPRSS2 effect. For that the authors should add condition when cells are transfected with ACE2+TMPRSS2.

In contrast to HeLa cells, Calu3 cells express endogenously TMPRSS2 and SKI-1. We did not succeed in efficiently transfecting Calu3 cells using various transfection reagents, thus we relied on the endogenous expression of TMPRSS2 in Calu3 cells. Furthermore, overexpression of TMPRSS2 was found to create active TMPRSS2 too early in the secretory pathway resulting in artificial activity of TMPRSS2 in the ER. This is why we eliminated all overexpression experiments of TMPRSS2 in our previous manuscript (see Ref 9) as they give erroneous results on the artificially induced early cleavage of S-glycoprotein in the ER, rather than the physiological cell-surface processing of S by TMPRSS2.  

5- In 3.2 the authors conclude that SKI1 increases fusion by acting on metalloprotease activities. However, in 3.3 the authors also investigate the effect of SREBP-2 and the mevalonate pathway and conclude that increase in cholesterol synthesis can also lead to fusion activation. If that is the case the entry of pseudotypes into Calu3 should be affected as well. Therefore, conclusions drawn by the authors are not really supported by the experiments. This should be investigated with some additional experiments or conclusions should be modified.

As shown in Figure 1A, in many cells including Calu3, pseudoparticles can enter the cells by either endocytosis or fusion upon Spike priming and activation. SKI-1, via cholesterol synthesis and metalloprotease activation affects only the entry of pseudoparticles by fusion, but not by the endocytosis pathway. Therefore, the fact that the mevalonate treatment did not significantly affect viral entry in Calu3 cells suggests that endogenous TMPRSS2 exerts a dominant cleavage at S2’, which is a more powerful activity than that generated by cell-surface endogenous ADAM10/17. In contrast, HeLa cells do not express TMPRSS2, likely providing a possible rationale for the enhanced cell-to-cell fusion effects of SKI-1, mevalonate and ADAMs.  

6- Experiments with PCSK9 and ACE2 should be repeated in a cell line with endogenous levels of ACE2 (like Calu3). In transfected cells levels of ACE2 are very high and I am not convinced that depletion of 30% shown in Figure 7D is significant. The fact that there is no LDLR depletion with the WT PCSK9 could also suggest that overexpression of ACE2 or LDLR is not a good system to show the real effects of PCSK9. The authors suggest in the discussion that the lack of LDLR degradation could be due to the presence of ACE2 but that should be supported by a control where only LDLR is expressed.

We agree with the referee that the ability of PCSK9 to degrade ACE2 should be validated in different cell types. Currently we are repeating the experiment, including the expression of endogenous LDLR alone, in  HEK293 and HepG2 cells, especially as the latter cell line does express endogenously ACE2 (PMID: 34354120). We therefore ask the indulgence of the referee to allow us in a future manuscript to properly define the subcellular pathways implicated in PCSK9-ACE2 interaction and their subsequent degradation in various cell types in the presence or absence of LDLR or even CAP1 and MHC-I proteins (see ref. 65).  

7- Increase in fusion with LDLR expression is very close to EV control and therefore I don’t think this is significant.

The difference in cell-to-cell fusion is indeed small between EV and LDLR expression conditions, but it is significant (P<7x10-5), which is why we did added (***) in the bar graphs of Supplementary Figure S3B. Furthermore, the decrease in cell-to-cell fusion by PCSK9 is very significant (***) both in the presence of absence of LDLR (Figure S3B).

8- In Figures 2 and 4 on S blots levels of S2” are rather small with ACE2 transfection and they increase with SKI1. But on Figures 6 and 7 it’s very high. How to explain that?

The difference of S2”levels between Figures 2/4 and 6/7 is due to the exposure time, when SKI-1 is expressed (Figures 2/4), the pictures were taken at low exposure to be able to correctly quantify the S2”in presence and absence of SKI-1. But in Figures 6/7 the pictures were taken after a long exposure to visualize the low levels of S2’.

We have added the following sentence in the legbends of Figure 6A and 7A :

Note that we had to overexpose the gel to better see S2’.

9- In the discussion the authors widely discuss data they obtained with a lot just being repeated from the results section.  It would be interesting to learn more about implications these results could have for human infection. For example, the omicron variant has been shown to be less reliant on TMPRSS2 than other variants. That usually results in endocytosis but maybe in some cell types SKI1 activation could increase plasma membrane fusion.

We agree and hence added the following sentence at the end of the discussion:

Finally, the cellular infection of SARS-CoV-2 while dependent on TMPRSS2 and furin activity, is quite different from that of SARS-CoV-2 bearing Omicron variants, as the viral entry of the latter seems to be independent of TMPRSS2 activity and/or levels [93-95]. It would be informative if such Omicron variants that could use the endocytosis pathway (Figure 1A) may also depend on SKI-1 activity for viral entry and/or replication, like other RNA viruses [5].

Minor points:

- In Calu3 cell fusion happens at the plasma membrane due to high levels of TMPRSS2 so labelling Figure 1A as Calu3 cell should be removed.

We agree, and Calu-3 annotation was removed from Figure 1A.

- Figure 1B shows furin as a transmembrane protein which is not the case.

Furin is indeed a transmembrane proprotein convertase. Please see Ref # 4, and the Uniprot entry annotation https://www.uniprot.org/uniprotkb/P09958/entry

- If P values are used it should be shown on graphs which conditions are compared.

The significant results compared are now indicated on the Figures 1C; 3A,B; 6B; S3A,B.

- Figure 2A, furin shown as part of spike? The schematic is misleading.

In the Figure 2A Furin was correctly positioned and replaced by Furin sites.

- Figure 2D wrong labelling for EV and Spike.

Thanks for the keen observation, we have now corrected the labeling of spike and EV in Figure 2D.

-Figure 3 data are shown as fold RLU over EV. Why is if different to other figures?

The legends in Figures 3A and B are now corrected: RLU instead Fold over EV.

- % of S2’ and S2” is shown for some lanes on Western Blots but not for others, why?

The % S2’ and S2” were now added in Figures 4C and 7A.

- On page 13, third paragraph, the authors talk about Figure 1A, B where is should be Supplementary Figure 1A, B.

For clarity, we have now replaced the sentence by:

Unexpectedly, LDLR alone can serve as a weak receptor for S-glycoprotein, as compared to ACE2 (Supplementary Figure S3A), and furin-cleavage at S1/S2 (Figure 2A) is also required for the LDLR-mediated fusion. Indeed,  the furin-resistant mutants mS1/S2 and mAS1/S2 [9] no longer permit LDLR-enhanced fusion (Supplementary Figure S3A).

- In the graphical model furin is again shown as transmembrane protein.

We respectfully disagree, as furin is indeed a transmembrane proprotein convertase. Please see Ref # 4, and the Uniprot entry annotation: https://www.uniprot.org/uniprotkb/P09958/entry

Round 2

Reviewer 1 Report

It should be much better when we see the mention or the discussion about  evidence on the relationship of viral infection and cholesterol biosynthesis  in vivo. period.

Author Response

The only comment of Referee # 1 now is:

"It should be much better when we see the mention or the discussion about  evidence on the relationship of viral infection and cholesterol biosynthesis  in vivo. period."

In the discusssion we added the sentence: "Furthermore, the correlation between LDL-cholesterol levels and COVID-19 disease severity is still unclear."

In that respect, the only in vivo work relating PCSK9 to COVID-19, just appeared during the revision of this paper on January 24, 2023 PMID: 36653090

In this work the authors concluded that :

    "PCSK9 inhibition compared with placebo reduced the primary endpoint of death or need for intubation and IL-6 levels in severe COVID-19. Patients with more intense inflammation at randomization had better survival with PCSK9 inhibition vs placebo, indicating that inflammatory intensity may drive therapeutic benefits. "

    Thus, aside from the ability of PCSK9 to promote the degradation of the LDLR(an hence reduce LDL-cholesteriol in vivo) our cell based studies showed that PCSK9 can also enhance the degradation of ACE2.

    In absence of PCSK9 higher levels of ACE2 may be protective against inflammation, as they will produce higher amounts of the antiinflammatory Angiotensin 1-7. Accordingly, we now have added the following sentences in the discussion supporting an in vivo role of PCSK9 in enhancing COVID-19 complications, including inflammation:

    "In that context, very recently it was reported that hospitalized COVID-19 patients receiving a single subcutaneous injection of the PCSK9-mAb evolocumab exhibited reduced death or need for intubation, as well as decreased inflammatory cytokine IL-6 levels in severe COVID-19 cases [92]. Our data in the present study and the outcomes of the above clinical trial suggest that lack of active circulating PCSK9 leading to reduced inflammation, as was reported before [33, 93-96], may in part be due to enhanced levels of ACE2 activity that would generate higher levels of the anti-inflammatory angiotensin 1-7 [97, 98]. Indeed, the potential of angiotensin 1-7 in the treatment of critically ill COVID-19 patients is now being evaluated in ongoing clinical trials [98]."

    Reviewer 2 Report

    The authors tried to answer my comments and where possible did some changes to the text and figures to improve the manuscript. They also provided good explanations to why some data was presented in one way or another.

    Although I asked for additional experiments that could confirm some of the results, with 10 days given by the editor for the revision I understand the authors would not be able to do that. 

    I still have few minor comments for the authors:

    - Spike protein schematic was modified for furin site in Figure 2, but not in Figure 4,

    - in Figure 6B the authors show some significance between conditions that look very similar. Is that right?

    - the authors added short paragraph about the Omicron variant, but this is placed after acknowledgement and not in the discussion.

    Author Response

    We thank the reviewer for his/her understanding and for the minor corrections proposed:

    1. In Figure 4, the term "furin sites" has now been incorporated, as requested

    2. Yes, they are similar but both are significantly different from the conftrol ACE2 alone. 

    3. We have now placed the sentence below at the end of the discussion as originally intended, so sorry for the error that is now corrected:

    "Finally, the cellular infection of SARS-CoV-2 while dependent on TMPRSS2 and furin activity, is quite different from that of SARS-CoV-2 bearing Omicron variants, as the viral entry of the latter seems to be independent of TMPRSS2 activity and/or levels [99-101]. It would be informative if such Omicron variants that could use the endocytosis pathway (Figure 1A) may also depend on SKI-1 activity for viral entry and/or replication, like other RNA viruses [5]."